# Women Taking a Folic Acid Supplement in Countries with Mandatory Food Fortification Programs May Be Exceeding the Upper Tolerable Limit of Folic Acid: A Systematic Review

**DOI:** 10.3390/nu14132715

**Published:** 2022-06-29

**Authors:** Carolyn Ledowsky, Abela Mahimbo, Vanessa Scarf, Amie Steel

**Affiliations:** 1School of Public Health, University of Technology, P.O. Box 123, Sydney, NSW 2007, Australia; carolyn.j.ledowsky@student.uts.edu.au (C.L.); abela.mahimbo@uts.edu.au (A.M.); 2School of Nursing and Midwifery, Faculty of Health, University of Technology, P.O. Box 123, Sydney, NSW 2007, Australia; vanessa.scarf@uts.edu.au

**Keywords:** folate, folic acid, 5-methyltetrahydrofolate, fortification, unmetabolized folic acid, preconception, pregnancy

## Abstract

Background: In preconception and pregnancy, women are encouraged to take folic acid-based supplements over and above food intake. The upper tolerable limit of folic acid is 1000 mcg per day; however, this level was determined to avoid masking a vitamin B12 deficiency and not based on folic acid bioavailability and metabolism. This review’s aim is to assess the total all-source intake of folate in women of childbearing age and in pregnancy in high-income countries with folate food fortification programs. Methods: A systematic search was conducted in five databases to find studies published since 1998 that reported folate and folic acid intake in countries with a mandatory fortification policy. Results: Women of childbearing age do not receive sufficient folate intake from food sources alone even when consuming fortified food products; however, almost all women taking a folic acid-based supplement exceed the upper tolerable limit of folic acid intake. Conclusions: Folic acid supplement recommendations and the upper tolerable limit of 1000 mcg set by policy makers warrant careful review in light of potential adverse effects of exceeding the upper tolerable limit on folic acid absorption and metabolism, and subsequent impacts on women’s health during their childbearing years.

## 1. Introduction

Governments worldwide encourage women of childbearing age to take folic acid (FA) prior to pregnancy to avoid neural tube defects (NTDs) [1,2,3,4,5,6,7,8,9,10] and some governments (e.g., Australia, Canada and United States) have implemented a FA food fortification program to ensure that women who unintentionally conceive or are not using a folic acid supplement have sufficient folate to prevent neural tube defects [11,12]. Since the mandatory folic acid fortification program began in 2009, Australian policy makers have recommended 400 mcg of FA in addition to fortification for individuals considering conception [6].

While the fortification policy has prevented NTDs, emerging research has documented the presence of unmetabolized folic acid (UMFA) in the serum in people in countries that have a FA fortification program and/or take a FA supplement [13,14,15]. While the effect of this is unclear, some have raised concerns regarding UMFA and adverse health effects [16,17,18,19,20,21].

### 1.1. Folate Metabolism

Folate-dependent biochemical processes are influenced by the form and bioavailability of folate [22]. Folate is a generic term for vitamin B9 that includes several forms, including that naturally found in food (natural folate). All folate forms have a common structure but differ based on the pteridine ring and whether it is reduced or oxidized [23]. Folate derived from food such as leafy green vegetables are polyglutamates (Figure 1), whereas synthetic FA is a monoglutamate (Figure 2) [21].

Food folate is naturally present in foods such as leafy green vegetables and is in a reduced form [13]. This natural folate exists in a polyglutamate form and needs to be further hydrolyzed to a monoglutamate form by the intestinal lumen to be transported [24]. The intestinal absorption of polyglutamate derivatives of tetrahydrofolate (THF) into monoglutamates is carried out by the brush border enzyme glutamate carboxypeptidase II (GPCII) [25]. This crossing of the apical brush border is carried out via two transporters, the proton-coupled folate transporter (PCFT) and the reduced folate carrier (RFC) [26,27]. Most of the absorbed natural folate is metabolized to 5-methyltetrahydrofolate (5-MTHF) in the intestine and/or liver [26] because the RFC has a higher affinity for reduced folates such as 5-MTHF compared to FA [26]. The fate of both natural folate and FA is to be metabolized to 5-MTHF [28]. The RFC transporter is found throughout the intestinal tract and is highly expressed in the liver and placenta [26]. The PCFT’s key role is as the major transporter of dietary folate in the proximal small intestine, specifically in the brush border of the duodenum and jejunum and has an optimal function with an acidic PH but has a higher affinity to FA than it does to reduced folates such as 5-MTHF [16,26,29,30]. Once absorbed by the RFC or PCFT, the folate is transported into the hepatic portal vein by multidrug resistance-associated protein 3 (MRP3) and then to the liver, where it is taken up via hepatocytes [26]. The intestine can convert reduced folates to 5-MTHF quite well; however, it is the first-pass metabolism and intestinal absorption where the difference between the metabolism of FA and the reduced folate forms exists as its reduction and methylation are dose dependent [14,28].

While natural folates do not need to be converted to 5-MTHF to cross the mucosal cell (Figure 3) and enter the portal vein to the blood, they do need to be metabolized to 5-MTHF in the liver or released into the blood or bile [27]. One small study showed that the majority of FA (at a physiologic dose) passes into the portal vein in an unmodified form, whilst monoglutamates are almost all converted to 5-MTHF [31]. All forms of folate must be converted to 5-methyltetrahydrofolate (5-MTHF), which is the main form of folate found in plasma, accounting for approximately 98% of total serum folate [16,23]. Where natural folates can be converted directly to 5-MTHF directly, FA relies on the dihydrofolate reductase (DHFR) enzyme to be reduced [21,24]. The activity of DHFR is slow and easily saturated in the first step conversion to dihydrofolate and therefore when the dose exceeds the capacity of the enzyme or there is a DHFR polymorphism, an increase in free FA in plasma may be seen [27]. The main circulating form of folate in the blood, 5-MTHF, is in high demand during fetal development as DNA synthesis and cell division are increased [26]. The folate receptor-α (encoded by the folate receptor 1 (FOLR1) gene) is expressed on epithelial cells and ensures uptake of 5-MTHF by the placenta and the brain during development. This receptor is upregulated in pregnancy due to the rapid increase in folate-dependent activities such as tissue growth, placental development, and enlargement of the uterus [26,32]. 5-MTHF donates its methyl group to support DNA methylation via DNA methyltransferase enzymes and 5-MTHF is directly linked to the viability of the embryo and the ability to maintain DNA methylation patterns in replicating cells [26,33]. A lack of 5-MTHF may therefore compromise DNA methylation, cause uracil misincorporation, chromosomal breaks, impaired ovarian follicle development and increased risk of pregnancy loss [32]. Folate metabolism can be impaired in people who carry the methylenetetrahydrofolate gene (MTHFR) polymorphisms, resulting in reduced enzyme activity and therefore a reduction in 5-MTHF [34,35,36]. The MTHFR gene has been linked to infertility in both men and women [37,38,39,40,41,42,43]. This is particularly pertinent given that studies in mice show that excess FA reduces MTHFR activity irrespective of MTHFR polymorphisms thereby reducing methylation capacity. Research has identified 5-MTHF as a possible alternative for FA in those with MTHFR polymorphisms as doses of 800 mcg have been shown to bypass the MTHFR enzyme [23,24,25,44,45,46,47].

### 1.2. Folic Acid Dose

Although mandatory fortification exists in some countries, the fortification levels per 100 mg food differ across each country, with no upper limit on the number of foods that may be fortified voluntarily, easily leading to an intake level above the original estimated level of 100 mcg per day [10,11,12,48,49,50,51,52,53].

FA is widely used in supplements and food fortification as it is relatively inexpensive, stable, and freely available [7,54,55,56]. Women are encouraged by governments to take FA supplements during preconception and pregnancy in addition to any folate they may consume through natural folate and fortified foods. Doses of 400–500 mcg are recommended and included in prenatal multivitamins by the Department of Health in Australia [57], whereas the United States, the Centers for Disease Control and Prevention recommends 800 mcg per day [5,6,7] and in Canada at least 1000 mcg of FA are typically found in a prenatal multivitamin [51,58,59].

Women at high risk of pregnancy loss or who had a previous pregnancy affected by a NTD, are routinely prescribed FA doses of 4–5 mg in Australia, Canada, and the United States [6,7,9,46]. FA is considered to be better absorbed and therefore a measurement, known as *total food folate daily folate equivalent (DFE)* (see Box 1), has been devised by the Institute of Medicine and is the recommended method to provide comparability between synthetic FA and natural food folates whereby the folate from fortified food is weighted 1.7-fold greater than naturally occurring folate [10]. For example, a serving of food from leafy greens may provide 100 mcg of folate, which equals 100 mcg DFE; however, 100 mcg of folic acid in a fortified food is estimated to be 170 mcg DFE.

Box 1Daily folate equivalent (DFE).Total food folate DFE = mcg natural food folate mcg + synthetic folic acid from fortified foods and supplements times a bioavailability factor of 1.7 = mcg DFE/day (natural food folate) + (folic acid mcg × 1.7) [1,2] or 1 DFE equals 1 mcg food folate, 0.6 mcg synthetic folic acid consumed with food or 0.5 mcg consumed on an empty stomach with fortified food [3,4,5].

### 1.3. Upper Tolerable Limit of Folic Acid

Unlike 5-MTHF (and natural folate), FA has an upper tolerable limit (UL), which the US’ Institute of Medicine sets at 1000 mcg due to the potential for high FA intake to mask a vitamin B12 deficiency in the user [10]. The UL does not include naturally occurring folate from food or 5-MTHF as there is no recognized UL for these forms [28]. The studies used to determine this upper limit do not consider the potential impact of elevated FA intake on folate metabolism [16]. Apart from masking a vitamin B12 deficiency, the only documented risk factor for excessive FA intake is the presence of unmetabolized folic acid (UMFA) [13,16,19,21,60,61]. UMFA is the FA that cannot be metabolized to 5-MTHF due to limited DHFR activity and, therefore, starts to appear in the circulation even with 200 mcg of FA acid intake per day [62,63]. This UMFA has been linked to adverse health effects including reduced brain function and motor and somatosensory processing in offspring [64], cleft lip [65], asthma [52,66], autism [67] anemia [68], natural killer cell cytotoxicity [61], adverse cardiac events and cancer [21] and cognitive impairment [68].

There is a risk that women may be exposed to unintended adverse health outcomes through excess FA intake; for this reason, it is imperative that preconception and pregnancy folate intake from all sources for women of childbearing age is more clearly understood.

## 2. Materials and Methods

This review aimed to assess the total all-source intake of folate in women of childbearing age and in pregnancy in high-income countries with folate food fortification programs.

This systematic review was designed in accordance with the Preferred Reporting Items for Systematic reviews and Meta-Analyses (PRISMA) guidelines.

We conducted a detailed search of the following electronic databases: MEDLINE (OVID), Embase, CINAHL (EBSCO), Scopus, and BiblioMAP. The search strategy is attached (Appendix A, Search Strategy) and was conducted on the seventh and eighth of October 2021. It included countries identified by the World Bank as high-income countries, pregnant women or women in preconception, folate exposure and source of folate.

Studies published from 1998 until the search date of 8 October 2021 were included. We wanted to include only studies that reflected total folate intake and therefore the start date of 1998 was selected as this is the date that mandatory FA fortification began in the U.S. and other countries around the world [10,63,69]. Scientific posters and conference abstracts were not included. There were no language limits. A PICO(T) format was used to formulate the search strategy and help design the research question. The population of interest (P) was identified as women of childbearing age either in preconception or pregnancy period from high-income countries (as identified by the World Bank [69] that have a mandatory fortification policy of fortifying foods with FA. The intervention/exposure (I) was folate intake of any form, that is dietary folate from non-fortified foods, dietary intake from fortified foods and/or supplements. A comparator (C) was not included. We excluded control studies and intervention studies except where the control group was representative of the study group and nutritional/supplemental data were available. The outcome (O) measured was intake of folate in any form in women of childbearing age, whether in preconception or pregnancy. This included folate from natural food sources such as leafy green vegetables, FA from fortified foods and all folate forms in supplementary dietary products and multivitamins.

Titles and abstracts were reviewed by one reviewer (CL). Manuscripts were included if they were from epidemiological observational studies that explicitly report on folate intake in women of reproductive age—specifically in preconception and pregnancy. There were no language restrictions. Studies were excluded if they did not report on folate intake (rather than folate status), included women who were breastfeeding and were not from countries with a mandatory fortification policy.

Full texts were reviewed and assessed based on the inclusion and exclusion criteria above. If full texts could not be accessed, we contacted the authors and/or publisher and if the full text could not be located they were excluded. Any uncertainty about the inclusion of a study was discussed by the other authors (AS, VS and AM) and consensus was gained.

Figure 4 presents the flowchart of study selection and inclusion. A total of 1885 studies were identified. After removing duplicates (*n* = 780) and excluded citations by title and abstract (*n* = 648), 457 full-text studies were assessed for eligibility. Of those, 421 were excluded because they were either not measuring folate intake, not from a country that had a mandatory fortification policy, not an academic paper, or had an incorrect study design or date range. This left a total of 36 studies for inclusion.

Critical appraisal of the studies was conducting using the Axis tool for cross-sectional studies (Appendix A). The studies were assessed for key issues such as suitability of the study to answer the research question, and possibility of bias being introduced into the study. All studies except five [2,3,66,71,72] detailed funding sources and sources of conflict.

All studies stated clear aims and objectives and the study design was appropriate for the aim.

The few areas that data were lacking was in the sample size justification with twenty [2,3,33,53,58,59,62,72,73,74,75,76,77,78,79,80,81,82,83,84] of the thirty-six studies giving data on sample size justification. Another question that was not answered by many studies was the area of non-responders; few studies (*n* = 9) [3,33,59,62,66,71,77,80,83] gave information about non-responders. Twenty one of the thirty six studies addressed and categorized non-responders. Clarity around statistical significance and/or precision estimates was also lacking.

## 3. Results

The 36 studies included in this review were separated into two categories, those sampling women of childbearing age (*n* = 18) and those sampling pregnant women (*n* = 21). Three studies were included in both categories. The study designs were predominantly cross-sectional or cohort and provided data about the folate intake from all sources in the sample populations. Our analysis of these studies identified four key areas of discussion or themes: natural folate intake, food fortification intake, supplement intake and the rate of women exceeding the upper tolerable limit of 1000 mcg FA.

### 3.1. Study Characteristics

#### 3.1.1. Women of Childbearing Age

Table 1 shows summarized data from the 18 studies sampling non-pregnant women of childbearing age [1,2,3,4,5,33,53,72,77,78,81,82,84,85,86,87,88,89]. The age of women included in the studies was 17–49 years of age and sample sizes ranged from 51 participants to 9707, with a total of 52,580 women being sampled across all studies. All studies were from the United States (*n* = 15), Canada (*n* = 2) and Australia (*n* = 1).

#### 3.1.2. Pregnant Women

Table 2 shows summarized data from 21 studies sampling pregnant women [50,51,52,58,59,62,66,71,73,74,75,76,78,79,80,83,88,90,91,92,93]. Two studies included women in their first trimester [71,75], nine studies included women in either their first or second trimester [50,59,62,66,74,76,80,92,93], one study included women with less than 26 weeks gestation [52], one study less than 27 weeks gestation [58], three studies in their second or third trimester [51,88,91] and five studies at any stage of gestation [73,78,79,83,90]. The sample size ranged from 51 participants to 2146, with a total of 16,314 sampled across all studies. All studies were from Canada (*n* = 8), United States (*n* = 7) or Australia (*n* = 7). One study was conducted before the mandatory folic acid fortification program commenced [76].

### 3.2. Natural Food Folate and Fortification

In women of childbearing age, the recommended intake of 400 mcg folate per day was not met by any study from natural folate sources (see Table 1). One U.S study found that fortified cereals and grains contributes approximately 70.8% of dietary folate [5]. The studies showed that the mean intake of folate from natural food folate ranged from 228.5 to 324.3 mcg/day after fortification [72], contributing to a 50% increase in serum folate concentrations and a 59% increase in RBC concentrations [53]. Another study showed that 43% of folate came from both natural and fortified foods [86]; however, food alone did not achieve the recommended 400 mcg/day for women in childbearing years. One study that investigated only food sources of folate found that 80% of women of reproductive age did not meet the recommended daily allowance (RDA) of 400 mcg per day [88], while another study showed that only 23% of women achieved the RDA of 400 mcg per day [83].

In Australia, two studies by Dorise and Beringer found that only 20% [74] and 55% [90] of pregnant women met the folate recommendations from diet alone (including natural food folate and fortified foods), respectively. In another study, folate was consistently below the estimated average requirement (EAR) of 520 mcg [78] particularly in the periconceptional period [80]. In Canada, 70% of women did not meet the EAR of 520 mcg of folate in first trimester of pregnancy [75] nor was the diet from food or fortification sufficient to achieve the minimum 400 mcg/day FA recommendation [50]. Food fortification and natural folates contributed approximately 303 mcg/day preconceptionally [92] and in pregnancy ranged from 282.2 mcg dietary folate equivalents (DFE)/day [83], 483 mcg DFE/day in early pregnancy [62] to 331 mcg/day in women 20–38 weeks gestation [88], and 465 mcg DFE/day in late pregnancy [62].

Table 3 outlines women’s folate intake by folate source or type. Of those studies that only measured natural food folate intake, women of childbearing age consumed between 170.92 and 259 mcg per day and pregnant women consumed between 140 and 483 mcg DFE. The average intake of folate from natural food folate was 205.48 mcg [1,4,5,33,89] in these studies.

Synthetic folic acids from fortified foods contributed 181.7 to 239 mcg per day in non-pregnant women and 96 to 768 mcg per day in pregnant women.

The combined food folate of natural and synthetic folic acid based on the conversion of folates to daily folate equivalents was as high as 500.5 mcg DFE (369.9 mcg) in women of preconception age [5] and 627.6 mcg DFE per day in pregnant women [83].

### 3.3. Supplementation and Upper Tolerable Limit

In women of childbearing age, supplements contributed 47.5% [53] to 57% of folate intake [85] and were the main reason why 2.4–7% of women exceeded the upper tolerable limit of 1000 mcg per day [4,53,84]. The intake of FA varied throughout the studies but was typically between 400 and 1000 mcg per day through the intake of supplements with folic acid. The percentage of women taking 400 mcg of FA per day ranged from 24% in one study [77] to approximately 78% in another [85], while women taking 1000 mcg of FA via supplements represented approximately 20% of the sample population [85,86].

The level of FA intake from vitamin supplements was much higher in pregnant women than in non-pregnant women due to the high rate of supplementation in pregnant populations. Specifically, FA supplements represented between 77% [79] and 92% [92] of FA intake in pregnant women with an estimation of supplementation contributing to 84% of FA intake in early pregnancy and 63% in late pregnancy [66]. While the intake of folic acid-based supplements was high in pregnant women, it was not in women in the preconception stage with one study reporting that 63% of women were not taking a prenatal with folate in the three months before falling pregnant [80].

In all studies reporting pregnant women exceeding the UL of folic acid, it was identified that FA-based supplements were the cause of their excess FA [50,52,58,59,66,73,75,76,79,80,93]. The percentage of women exceeding this UL ranged from 25% [58], 33.4% [73], 40% [79], 83–85% [59,62], 87% [75], 96% [50] to 100% [76]. It was estimated that supplements contributed to a mean daily intake of 878 mcg FA per day [79], 1000 mcg/day [59] to over 2000 mcg/day [50,66]. In one study, this overconsumption was more likely to occur in the first trimester [80].

It was difficult to evaluate a consistent method of overall intake of folate (food folate, fortified food, and supplements) as not all studies provide intake in DFEs and therefore only DFE levels will be reported. In women of childbearing age the total level of intake ranged from 864 mcg DFE [1] to 1778 DFE [86]. In pregnant women, total intake was much higher, ranging from 1451 mcg DFE [73] to 2181 mcg DFE [75]. The total DFE intake in pregnant women was recorded predominantly in Trimester 1 of pregnancy [50,62].

The overall intake of FA throughout pregnancy including supplements, not only caused an increased red blood cell folate [51,76] but contributed to total folate intakes up to 200% above the RDA [58], sometimes with a FA intake of up to 2948 mcg/day [66]. The studies measuring UMFA showed that it was measurable in all pregnant women [51] and even early first-trimester maternal plasma UMFA was detected in 97% of women and 93% of cord blood samples [62].

## 4. Discussion

### 4.1. Natural Food Intake and Fortification

There is a general consensus that women of childbearing age require a minimum of 400 mcg of FA per day in addition to their normal dietary intake to reduce the risk of neural tube defects [10] and, based on studies such as the ones included in this review, it is clear that women of childbearing age, are not receiving sufficient dietary folate to protect against neural tube defects, but rather they are receiving on average, half the recommended dose of folate from food. In response to studies that supported this finding of low folate intake from the diet, mandatory fortification of food products was introduced in the United States in 1998, at a level of 140 mcg per 100 g in all cereal grain products [11]. In Australia, mandatory fortification in wheat flour commenced in 2009 with 200 mcg per 100 g of wheat flour [94]. This regulation was based on modelling studies that showed that the increase in FA was expected to be approximately 100 mcg of additional FA per day; however, it is clear from the studies in this review that the intake of FA from foods alone is closer to 239 mcg or DFE 582, with the combined folate intake from natural food folate and FA fortification reaching close to the required intake of 400 mcg in women of childbearing age 500 mcg DFE. The fortification of folic acid was introduced to prevent neural tube defects in populations that were vulnerable to low levels of folate and by all accounts FA has been successful in achieving this [95,96,97,98,99,100]. The fortification program, however, did not cap the number of foods being fortified, nor did it consider populations that may be vulnerable to large amounts of FA like children and older adults [21,30,101] or tailor the intake levels via supplementation suggested for women in preconception and pregnancy.

### 4.2. Folate Measurement

The assessment of folate intake from food (natural folate and fortified foods) in the studies included in this review is based on the assumption that FA has better bioavailability than natural folates. However, these assumptions were arguably based on limited research do not align with emerging evidence. Natural folates are in a reduced form and are assumed to have nearly half the bioavailability of foods fortified with FA (synthetic FA fortified foods) [10]. This assumption is based on one study [102] with only ten female participants [10,102]. Researchers have since suggested that this finding may be incorrect [30] with studies investigating the bioavailability of natural folate to be anything from 78% of that of FA in food [103], and 98% for other foods [104]. A very detailed report on folate chemistry and metabolism suggests that folate polyglutamates (natural food folate) are not less bioavailable than monoglutamates (synthetic folic acid) [27]. DFE assumes that synthetic FA has a higher bioavailability compared with natural folate [33]. The measurement of food folate is difficult as confounding factors like loss of folate due to cooking, brush border enzyme activity, pH of the intestinal lumen, certain nutrients affecting absorption, and poor compliance in studies where participants are faced with difficult interventions [27,104,105,106], contribute to the accuracy of the study. It is important, therefore, that we gain a better understanding of folate metabolism because if we are underestimating the amount of natural folate that is bioavailable, this may have a bearing on FA fortification rates currently and future directions for many governments around the world. Further bioavailability research should also compare FA to the more active form, 5-MTHF, given studies have shown that bioavailability of both folinic acid and 5-MTHF are equivalent to FA as measured by changes in plasma folate levels [30,107].

### 4.3. Folate Metabolism

This review has highlighted the extent to which women of childbearing age, particularly in pregnancy, are exceeding the UL of FA, and this may be impacting on their folate metabolism. Bioavailability is very different to metabolism; although FA may be more bioavailable, it is not metabolized the same was as natural folate. Natural folate uptake by the brush boarder enzymes does depend on the type of food, the pH of intestinal tract and quality of the mucosa; however, there is no solid evidence that it is 50% less absorbed than FA and the studies show huge variation in percentage uptake of natural folates versus FA. Bioavailability is key because if FA is not metabolized sufficiently to 5-MTHF, which is the main circulating form of folate used by the placenta and brain, then the increased amount may be inhibiting the production of 5-MTHF and its re-uptake. It has been found that synthetic FA has no coenzyme activity until it can be reduced by dihydrofolate reductase (DHFR) [28] and that the DHFR enzyme is easily saturated which means that above a dose of 200–300 mcg the normal intestinal absorption of is saturated and rather than being converted by DHFR to dihydrofolate (DHF) or tetrahydrofolate (THF), we begin to see an accumulation of unmetabolized folic acid in the serum [14,62,108,109]. This then becomes an important question regarding both fortification and supplementation. If as studies in this review suggest, that the level of FA from fortification alone is reaching this saturation point for FA metabolism, and women, particularly in pregnancy are almost always exceeding this threshold, should the recommendation of further FA in preconception and pregnancy be changed to 5-MTHF to achieve the recommended folate level? 5-MTHF cannot accumulate [26,27] and is considered to be as good as FA, if not better in raising folate levels and preventing neural tube defects [23,24,25,46,110,111]. The studies in this review confirm that food-based folate is not sufficient to achieve recommended levels of folate in preconception and pregnancy; however, they do highlight that although FA may be considered to be absorbed better, it is not metabolized into 5-MTHF like natural folates and is accumulating instead.

Studies that have investigated 5-MTHF as a possible substitute for FA found that 5-MTHF at a dose of 7.5 mg every 12 h over four days rapidly increased serum total folate and has an advantage over FA as it can replete body stores in folate insufficient women within a few days [110]. 5-MTHF compared to folic acid was found to increase plasma folate levels more effectively [111]. Safety studies of high-dose 5-MTHF in rats showed no adverse events [112] and dosages as high as 17 mg of 5-MTHF daily for 12 weeks have been shown to have no side effects [113]. In pregnancy, there have been reported no side effects with 1.13 mg [114] and 7.5 mg and 15 mg 5-MTHF have been used without side effects in patients with depression [115,116]. 5-MTHF is more effective at improving folate status and may therefore be a better alternative to folic acid in those with infertility and recurrent pregnancy loss [24,46].

### 4.4. Unmetabolized Folic Acid

The fact that previous studies show people exposed to 200 mcg per day or less of FA are unlikely to have unmetabolized folic acid in serum, yet 300 to 400 mcg or above could lead to UMFA [16,109], suggest that all the women in the studies in this review in preconception or pregnancy would have levels of UMFA in their serum. Murphy et al. in their study, showed that all the women had the presence of UMFA irrespective of the dose [51] and Plumptre et al. found that 93% of cord blood samples had UMFA detected [62]. UMFA has been implicated in reduced natural killer cell activity in post-menopausal women [61], various forms of cancer, cardiovascular disease [21], autism [117] and orofacial clefts [118,119]. This has serious implications, given the studies in this review show that many women in preconception and almost all women in pregnancy have the presence of UMFA and are exceeding 400 mcg of FA per day. UMFA has also been found to cause a pseudo-MTHFR deficiency in rats [120] and reduce 5-MTHF. This effect has concerning implications as low folate is associated with uracil misincorporation [121], reduced DNA methylation [120] and changes in epigenetic patterns [122]. So too, the presence of UMFA has been linked to aberrant DNA methylation [123]. Methylation and the one carbon metabolism cycle (Figure 5) is a critical pathway required for oogenesis [124] and spermatogenesis [125,126] and is required for epigenesis and imprinting [127] and folate is required for the synthesis of purines and thymidylate for RNA and DNA synthesis [28], thus having implications for preconception, pregnancy, birth defects and the outcome of the offspring. The folic acid UTL of 1000 mcg has been set based on the potential for high folic acid intake to mask a vitamin B12 deficiency [16,28,62] and not the level at which serum UMFA appears. Based on the studies in this review, every single woman may exceed the level of FA intake where UMFA would appear. This is not the case with 5-MTHF which cannot accumulate and has a feedback loop to ensure that there is a homeostatic mechanism that prevents excessive levels of folate in tissues [28]. Previous research has shown that high-dose folic acid results in a global loss of methylation across the sperm methylome and leads to altered sperm epigenome and sperm DNA methylation, lower pregnancy rates, infertility rates and abnormal embryos [128,129].

A lack of 5-MTHF contributes to an elevation in follicular levels of homocysteine [35,130] which correlates to adverse pregnancy outcomes as folate receptor-α has a high affinity for 5-MTHF and ensures cellular uptake where the folate receptor is expressed in the ovary, oocytes and human embryonic stem cells [32]. These biochemical mechanisms may underpin the reported links between MTHFR polymorphism and infertility [37,38,40,41,131], implantation failure [22] and recurrent pregnancy loss [24,36,46,130,132]. Such mechanisms may also explain preliminary research that has found replacing FA supplements with folinic acid or MTHF supplements may increase pregnancy success in women with MTHFR polymorphisms who have experienced recurrent pregnancy loss [133].

### 4.5. Upper Tolerable Limit

The studies in this review confirm that many women preconceptionally and almost all women at some stage in pregnancy are exceeding the UL of 1000 mcg/day. The upper limit seems to be almost inconsequential as it is simply a consideration of vitamin B12 masking and does not take into account the consideration of metabolism. If metabolism is a concern, then all women in pregnancy are exceeding the threshold and most women taking a prenatal multivitamin most certainly are as well. Given the documented concerns regarding the presence of UMFA, this upper tolerable limit needs to be re-evaluated.

## 5. Conclusions

Natural folates may not deliver the optimal amount of folate to reduce the risk of NTDs or methylate DNA sufficiently to prevent miscarriage. Women may be at risk of producing UMFA due to their intake of FA from food fortification. Any supplementation of FA over and above that has the potential to cause adverse health outcomes, reduce MTHFR activity, mask vitamin B12 deficiency, reduce methylation of DNA and therefore contribute to adverse pregnancy outcomes such as pregnancy loss. The results of this review warrant close attention from public health researchers and policy makers and may justify a review of the current approach to ensure folate sufficiency in the general population and avoidance of UMFA levels that may influence fertility and pregnancy outcomes.

## Figures and Tables

**Figure 1 nutrients-14-02715-f001:**
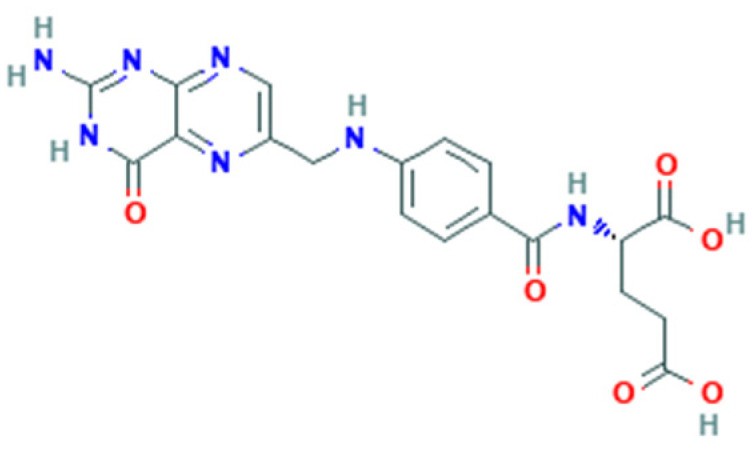
Folic acid structure.

**Figure 2 nutrients-14-02715-f002:**
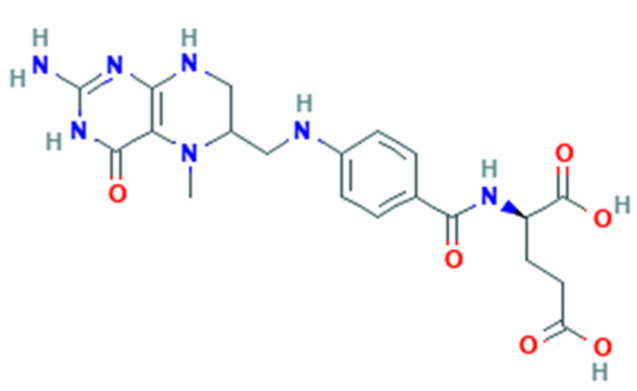
5-MTHF structure.

**Figure 3 nutrients-14-02715-f003:**
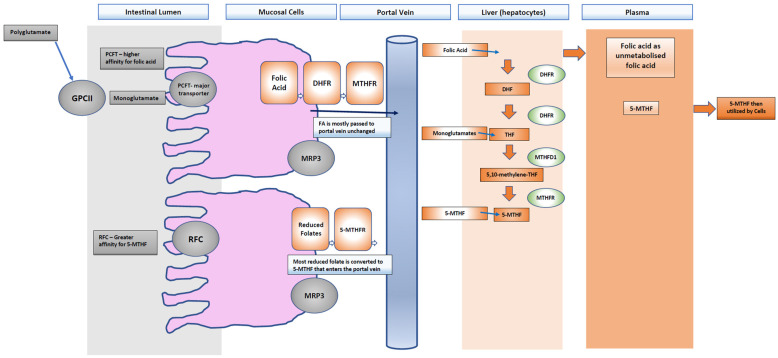
Polyglutamates are converted to monoglutamates by glutamate carboxypeptidase II GPCII and absorbed by the reduced folate carrier (RFC) or the proton-coupled folate transporter (PCFT). Almost all the reduced folates are converted into 5-MTHF and transported to the portal vein by multidrug resistance-associated protein 3 (MRP3) to the liver and blood. Folic acid is mostly passed unchanged to the portal vein, where it needs to be metabolized in the liver to dihydrofolate (DHF) by the enzyme dihydrofolatereductase (DHFR) and to tetrahydrofolate (THF) also by DHFR and finally to 5-methyltetrahdyrofolate (5-MTHF). What is not converted will pass into the plasma.

**Figure 4 nutrients-14-02715-f004:**
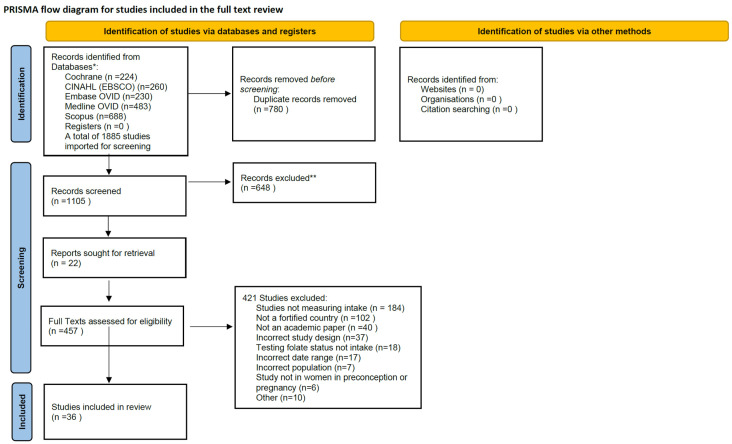
PRISMA-P flowchart of study selection [70]. Quality Assessment of Study. * *p* < 0.01; ** *p* < 0.001.

**Figure 5 nutrients-14-02715-f005:**
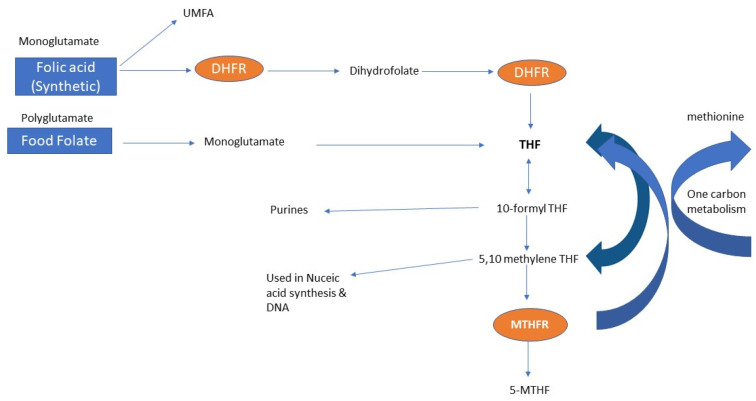
Folate metabolism leading to one carbon metabolism—the end goal is for folates to be metabolized to 5-MTHF, where it is used by methionine synthase reductase (MTR) for one carbon metabolism and methylation reactions: 5-MTHF = 5-methyltetrahydrofolate reductase, DHFR = dihydrofolate reductase, THF = tetrahydrofolate, MTHFR: methylenetetrahydrofolate reductase, and UMFA: unmetabolized folic acid.

**Table 1 nutrients-14-02715-t001:** Included studies sampling women of childbearing age, not pregnant.

Author/ Year	Location	Study Design	Study Period	Population Description	Total Number of Participants	Objective of the Study	Summary of Key Findings in Relation to Folate
Cena 2008 [1]	United States	Cross-sectional study	Prior to 2008	Women 18–45 years	157	To assess folate intake among low-income, non-pregnant women of childbearing age	* ~85% met RDA for folate * 4 exceeded UL of 1000 mcg FA/day food, fortified foods and supplements
Crider 2018 [33]	United States	Cross-sectional study	2007–2012	Women 12–49 years	4783	To estimate the usual daily FA and RBC to prevent NTDs	* If women only have fortified foods > risk of NTDs. * Require additional FA intake to achieve 400 mcg/day
Dietrich 2005 [3]	United States	Cross-sectional study	1999–2000	Women 20–39 years	2260 NHANES III 356 NHANES 1999–2000	To explore the changes in serum and erythrocyte folate following FA fortification	* Fortification increased the serum folate levels to acceptable levels * <10% reach RBC folate to reduce NTD risk
French 2003 [4]	Canada	Cross-sectional study	2001–2001	Women 18–45 years	148	To estimate folate intake and knowledge in women of childbearing age, in relation to risk of NTDs	* 7% exceeded the UL * Fortification does not cause > UL, supplements do
Gaskins 2012 [5]	United States	Cohort study	2005–2007	Women 18–44 years	259	To evaluate the association between dietary FA intake and hormones in healthy, women	* Mean dietary intake of 500 mcg achieved without supplementation (FA 50.8%/49.2% natural folate * 29.1% of dietary folate came from fortified cereals 41.1% from fortified grains * 18.1% from vegetables and 11.7% from beans.
Gaskins 2014 [85]	United States	Cohort study	2006–2013	Women 18–46 years	316	To evaluate the association of folate with ART outcomes	* ~57% women’s folate came from supplements * 78% took supplements with 400 mcg FA, 19% 1000 mcg * 43% from foods (both natural and fortified)
Gaskins 2019 [86]	United States	Cohort study	2004–2017	Women 18–46 years	513 ART cycles from 304 women.	To evaluate folate intake air pollution and livebirth in women using ART	* 20% of women consumed 1000 mcg of supplemental FA
Hamner 2013 [77]	United States	Cohort study	2001–2008	Women 15–44 years	5369	FA fortification to increase folate levels in Mexican women with lower acculturation	* 24.0% women had FA intake of 400 mcg
Hure 2008 [78]	Australia	Cross-sectional study	2003–2003	Women 25–30 years	9076 (606 pregnant)	To investigate and report the diet quality of young Australian women by pregnancy status	* Folate < EAR * Dietary deficit * No supplements recorded
Jun 2020 [79]	United States	Cross-sectional study	1999–2014	Women 20–44 years	8096	To estimate dietary supplements use and prevalence	* 44.8% used dietary supplements * Only supplements measured
Marchetta 2016 [81]	United States	Cross-sectional study	2001–2010	Women 15–44 years	4985 NHW and MA non-pregnant women	To assess the differences in serum and RBC folate concentrations by acculturation factors	* Supplements with FA impact blood folate status * Acculturation factors impact folate levels * RBC folate concentrations indicate long-term status
Mojtabai 2004 [72]	United States	Cohort study	1999–2000	Women 17–49 years	1351	To discover the impact of BMI on serum folate levels	* Fortification increased FA levels from mean 228.5 to 324.3 mcg/day * Serum folate levels increased due to FDA mandate of fortification.
Pick 2005 [88]	Canada	Case control study	Prior to 2004	Women 20–40 years	112	To examine the diets of healthy women	* FA + natural folate in food does not achieve recommended 400 mcg/day * 80% of women did not meet the RDA for folate * Supplements not included
Rai 2014 [82]	United States	Cohort study	2003–2008	Women 19–50 years	3641	To evaluate nutritional status in women of childbearing age and ethnicity	* RBC folate cut off for deficiency is 200 nmol/L * Most women achieved minimum in relation to folate deficiency but few reached 906 nmol/L for prevention of NTDs * RDA of 400 mcg folate 90% did not achieve RBC folate levels to prevent NTDs * At 1000 mcg/d, less than 25% of women had RBC folate level that prevent NTDs
Sotres-Alvarez 2012 [89]	United States	Case control study	1997–2005	Women childbearing age	1047 cases with NTDs 6641 CHD 6123 nonmalformed controls	Dietary intake and NTDs and CHDs	* Non-users of FA/multivitamin supplements who ate more fruits and vegetables significantly less likely to have NTDs * Supplements not included
Tinker 2012 [84]	United States	Cohort study	2003–2008	Women 15 to 44 years	4272	BMI supplemental FA intake and folate status	* Women 25–44 years age more likely to use FA supplements than 15–24 year olds * Nonusers and users of dietary supplements similar FA intakes of DFEs * BMI may affect body distribution of folate
Tinker 2012 [2]	United States	Cross-sectional study	2003–2004, 2005–2006, 2007–2008 NHANES-2008	Women childbearing age	4272	We sought to model FA intake under various fortification and supplementation scenarios	* UL influenced by supplements * Median intake influenced by enriched cereal grain products * 23% of women achieve RDA of FA * 2.4% exceed the UL
Yang 2007 [53]	United States	Cohort study	2001–2002	Women 14–49 years	1685	To examine FA intake in women of childbearing age in the United States	* Average serum folate concentrations show 50% increase since fortification * RBC concentrations show 59% increase since fortification * 47.5% of FA intake from supplements * 5.7% women consuming supplements and fortified foods exceeded the UL * Underreporting of FA from fortified foods

ART = assisted reproductive cycles, BMI = body mass index, CHD = congenital heart defects, DFE = daily folate equivalent, EAR = estimated average requirement, FA = folic acid, NTDs = neural tube defects, RBC = red blood cell, RDA = recommended daily intake, UL = upper tolerable limit of folate (1000 mcg), and RBC = red cell folate.

**Table 2 nutrients-14-02715-t002:** Included studies sampling pregnant women.

Author/ Year	Location	Study Design	Study Period	Population Description	Total Number of Participants	Objective of the Study	Summary of Key Findings in Relation to Folate
Bailey 2019 [73]	United States	Cross-sectional study	2001–2014	Pregnant women 20–40 years at all stages of gestation	1003	To estimate nutrient intakes (from foods and dietary supplements) and prevalence of meeting or exceeding RDA among pregnant U.S. women	* Supplements responsible for UL being exceeded * Supplements required for pregnant women to meet RDA * 33.4% exceed UL FA
Beringer 2021 [74]	Australia	Cross-sectional study	2009–2019	Pregnant Indigenous women in all stages of gestation	152	To determine sources of key nutrients contributing towards nutritional adequacy during pregnancy	* 75% met EAR * 55% met folate EAR form diet alone * ~51% of women took a supplement containing folate
Boeke 2013 [71]	United States	Cohort study	1999–2006	Pregnant women in T1/T2	1896	To examine maternal T1/T2 dietary intake of methyl donor nutrients during pregnancy in relation to child visual memory	* Second trimester folate intake average was 1268 mcg/day
Dorise 2020 [74]	Australia	retrospective study	2015–2015	10–18 weeks gestation	231	To evaluate the effectiveness of a group-based outpatient dietary intervention in pregnancy to reduce excessive gestational weight gain	* Supplement required to meet EAR * 20% met with folate EAR from diet alone * Mean folate intake 533 mcg/day NB: measuring food (adjusted for supplementation)
Dubois 2017 [75]	Canada	Cohort study	2010–2012	Pregnant women T1	1533	To assess nutritional intakes during pregnancy by examining dietary sources and supplements and comparing to RDI	* 70% did not meet EAR with diet * With supplements 87% exceeded the UL (diet and supplement sources)
Furness 2013 [76]	Australia	Prospective observational study	Prior to 2009	Pregnant women < 20 weeks gestation	46 low risk and 91 high risk women = total of 137	To determine if methyl donor nutrients < 18–20 weeks gestation are associated with subsequent adverse pregnancy outcomes. To investigate maternal B vitamin concentrations with DNA damage markers	* Older women had increased RBC folate, serum folate * FA supplementation > 1000 mg/day resulted in the highest RBC folate * 100% of the high risk women exceeded the UL of FA (2116 mcg mean) range 1615–2617 mcg NB: study conducted before mandatory FA fortification from 18 September 2009 in Australia
Gomez 2015 [58]	Canada	Cohort study	2009–2010	Pregnant women < 27 weeks gestation	599	To describe the use of natural health products (NHP) by pregnant women in each trimester of pregnancy	* Average FA intake was 200% above RDA in each trimester * 25% pregnant women exceeded UL in each trimester * IOM guidelines met by 97% in first trimester, decreasing to 91% in third trimester NB: supplement use only recorded
Hromi-Fiedler 2011 [91]	United States	Cross-sectional study	2004–2006	Pregnant Latinas between 16 and 32 weeks gestation	241	To document nutrient and food intakes from food sources among Latina subgroups living in the same geographical area	* Mean FA intake 768 mcg from diet alone NB supplement use not recorded
Hure 2008 [78]	Australia	Cross-sectional study	2003–2003	Women aged 25–30 years at any stage of pregnancy	606	To investigate and report the diet quality of young Australian women by pregnancy status	* Folate was consistently below EAR * Only food reported/no supplements
Jun 2020 [79]	United States	Cross-sectional study	1999–2014	1314 women 20–44 years of age at all stages of pregnancy	1314	To estimate the prevalence of use and the micronutrient contribution of dietary supplements among pregnant, lactating, and non-pregnant and non-lactating women	* 77% used dietary supplements * More than 60% of pregnant women used supplements with FA * Mean intakes of FA from supplements alone were at or above the RDI * Dietary supplements contributed a mean daily intake of 787 mcg of FA * >40% supplement users exceeded the UL of FA
Livock 2016 [80]	Australia	Cohort study	2011–2012	Women < 19 weeks gestation	2146	To examine overall micronutrient intake periconceptionally and throughout pregnancy	* Many women failed to meet RDI for folate in periconceptional period. (fortified food not included) * UL exceeded through diet and supplemental sources esp. late in T1 * 80% of folate for overconsumers coming from supplements NB: food fortification not included
Martinussen 2011 [92]	United States	Cohort study	1997–2000	Pregnant women < 24 weeks gestational age	1499	To assess whether FA intake during T1 of pregnancy is related to asthma in the offspring by the age of 6 years	* Mean intake FA ↑ from 303 mcg preconceptionally, to 404 mcg in 1st, 605 mcg in 2nd and 676 mcg in the 3rd month of pregnancy * Mean supplementation in T1 was 497 mcg * 51% women used FA supplements before pregnancy * 61% used FA supplements in first month of pregnancy * 81% in second and 88% in third month of pregnancy 92% women used FA in first trimester > 800 mcg FA NB: only supplements measured. No food or fortification
Masih 2015 [59]	Canada	Cohort study	2010–2012	pregnant women at < 16 weeks gestation	353	To determine dietary and supplemental intakes and major dietary sources of one-carbon nutrients	* 85% women exceeded UL FA through supplements alone * Typical dose FA in supplements 1000 mcg * Pregnant women exposed to FA 2.5-fold RDI of 400 mcg/day
Murphy 2021 [51]	Canada	Case control study	2013–2015	Pregnant women 24–26 weeks gestational age 18–44 years of age	51	This was an ancillary study within the Folic Acid Clinical Trial (FACT), a randomized, double-blinded, placebo-controlled, phase III trial designed to assess the efficacy of high-dose FA to prevent preeclampsia	* All women exceeded the WHO RBC total folate 906 nmol/L cut off for NTDs * ~80% of women were above the 97th percentile for RBC folate concentrations * Folate status for all women > WHO cut off for NTD risk reduction * UMFA measurable in all women and some at high levels * High-dose FA is unwarranted for this clinical population
Rose/Murphy 2021 [50]	Canada	Case control study	2011–2015	Pregnant women 8–16 weeks gestation	1198	To evaluate the dietary and supplemental intakes of FA and to determine the proportions of pregnant women exceeding the estimated average requirement (EAR) and tolerable upper intake level (UL)	* FA intake from diet (food and fortification) insufficient to achieve 400 mcg FA/day * Median food/fortification 333 mcg DFE/day * If FA supplements are added mean folate 2167 mcg DFE/day * 89.2% of participants > EAR if supplements taken * 96% exceeded UL 1000 mcg/day, 0.4% below RDA (400 mcg)
Pick 2005 [88]	Canada	Case control study	Prior 2004	Women 20–38 weeks gestation aged 20–40 years	112	The objectives of this pilot study were to examine the diets of pregnant women and healthy women of child-bearing age	* Daily dietary folate intake for pregnant women was 331 mcg/day * 98% of pregnant women did not meet minimum RDI for folate from food Note: Supplements not included in study
Plumptre 2015 [62]	Canada	Cohort study	2010–2012	Pregnant women aged 18–45 yers between 10 and 22 weeks gestation	368	Determine maternal and cord blood concentrations of folate and unmetabolized folic acid (UMFA) and examine effect of maternal intakes of folate and FA and fetal genetic variants in folate metabolism on folate status	* Folate intake (natural folate and fortified foods) mean 483 mcg DFE/day early T1 and 465 mcg DFE/day late preg * 83% > UL FA * Median maternal folic acid intake was 1000 mcg early T1 * Early T1, maternal plasma UMFA detected in 97% of women * Maternal serum folate concentrations significantly decreased during pregnancy, whereas RBC folate significantly ↑ * UMFA detectable in 93% of cord blood samples
Roy 2012 [93]	Canada	Cohort study	2002–2005	Pregnant women between 10 and 22 weeks gestation	2019	Examine dietary intake of iron, zinc and folate, from food and supplement sources	* Mean food intake of FA 473 mcgDFE/day * Mean FA supplement intake 1338 mcg DFE/day, mean total folate 1811 mcg DFE/day * Only 16% did not reach RDA of 600 mcg DFE/day * Supplement intake > UL 1000 mcg
Shin 2016 [83]	United States	Cross-sectional study	2003–2012	795 pregnant women at all stages of gestation	856	To examine relationship between pre-pregnancy weight status/diet quality and nutritional status	* Normal-weight women had a mean dietary intake of 282.2 mcgDFE/day, and dietary supplemental intake 1329 mcgDFE/day
Trivedi 2018 [52]	United States	Cohort study	1999–2002	Pregnant women 1–26 weeks gestation—T1 and T2	1279 mother-child pairs	To examine this association in the United States, where the food supply is generally fortified with FA	* Mean intake of FA first trimester 930 mcg, second trimester 1238 mcg * ~75% exceeded 400 mcg * 94% women took FA supplement
Whitrow 2009 [66]	Australia	Cohort study	1998–2005	Pregnant women < 16 weeks gestation	557	To investigate the effect of the timing, dose, and source of folate during pregnancy on childhood asthma	* Median intake of FA from supplements was 2948 mcg/day * Supplements contributed to 84% of FA in early pregnancy and 63% in late pregnancy NB: Study conducted before fortification in Australia implemented

DFE = daily folate equivalent, EAR = estimated average requirement, FA = folic acid, NTDs = neural tube defects, IOM = Institute of Medicine, RBC = red blood cell, RDA = recommended daily allowance, RDI = recommended dietary intake, T1 = Trimester 1, T2 = Trimester 2, T3 = Trimester 3, UL = upper tolerable limit of folate 1000 mcg/day, UMFA = unmetabolized folic acid, and WHO = World Health Organization.

**Table 3 nutrients-14-02715-t003:** Folate intake per day by natural food intake, fortified food intake, supplements and total.

Author/ Year	Natural Food Folate Intake Dietary	Food Folate and FA Fortified Foods Combined Intake	Synthetic FA Intake—from Fortified Foods	Synthetic FA Intake from Supplements	Folate Intake Total (Food and Supplements)	Supplement Use during Study	Stage of Pregnancy/ Weeks Gestation
Women of childbearing age/Not pregnant							
Cena 2008 [1]	180.7 mcg	n/a	253.4 mcg	148.6 mcg	402.0 mcg, 864.0 mcg DFE	yes	NIL
Crider 2018 [33]	236 mcg	n/a	239 mcg, 582 mcg DFE	461 mcg	661 mcg, 1341 mcg DFE	yes	NIL
Dietrich 2005 [3]	n/a	294 mcg DFE	n/a	n/a	n/a	no	NIL
French 2003 [4]	259 mcg DFE	n/a	470 mcg DFE	n/a	812 mcg DFE	yes	NIL
Gaskins 2012 [5]	181.7 mcg	500.5 mcg, 368.9 mcg DFE	181.7 mcg	n/a	n/a	no	NIL
Gaskins 2014 [87]	n/a	764.54 mcg DFE	n/a	1013.46 mcg	1778 mcg DFE	yes	NIL
Gaskins 2019 [86]	n/a	459.1 mcg	n/a	338.4 mcg	797.5 mcg	yes	NIL
Hamner 2013 [77]	n/a	244 mcg	n/a	380 mcg	n/a	yes	NIL
Hure 2008 [78]	n/a	265.9 mcg	n/a	n/a	n/a	no	NIL
Jun 2020 [79]	n/a	n/a	n/a	375 mcg	n/a	yes, no food	NIL
Marchetta 2016 [81]	n/a	n/a	n/a	381 mcg	n/a	yes	NIL
Mojtabai 2004 [72]	n/a	332.1 mcg	n/a	n/a	n/a	no	NIL
Pick 2005 [88]	n/a	300 mcg	n/a	n/a	n/a	no	NIL
Rai 2015 [82]	n/a	456.4 mcg DFE	n/a	n/a	n/a	no	NIL
Sotres-Alvarez 2013 [89]	170.92 mcg, 430.70 mcg DFE	430.70 mcg DFE	130.76 mcg DFE	n/a	n/a	no	NIL
Tinker 2012 [2]	n/a	434 mcg DFE	n/a	>400 mcg, 475 mcg DFE	n/a	yes	NIL
Tinker 2012 [84]	n/a	n/a	n/a	n/a	n/a	no	NIL
Yang 2007 [53]	151 mcg	221 mcg	128 mcg	n/a	n/a	no	NIL
Pregnancy Studies							
Bailey 2019 [73]	n/a	630 mcg DFE	n/a	n/a	1451 mcg DFE	yes	All
Beringer 2021 [90]	n/a	502.6 mcg DFE	n/a	833.4 mcg DFE	996.6 mcg DFE	yes	All
Boeke 2013 [71]	n/a	n/a	n/a	n/a	T1 972 mcg T2 1268 mcg	yes	T1
Dorise 2020 [74]	n/a	533 mcg	n/a	n/a	n/a	no	10–22 weeks
Dubois 2017 [75]	n/a	463 mcg	n/a	n/a	2181 mcg DFE	no	T1
Furness 2013 [76]	283 mcg	n/a	n/a	668–2116 mcg	n/a	yes	18–20
Gomez 2015 [58]	n/a	n/a	n/a	T1: 1225 mcg T2: 1353 mcg T3: 1228 mcg	n/a	yes, no food	<27
Hromi-Fiedler 2011 [91]	284.9 mcg	610 mcg	768 mcg	n/a	n/a	no	16–32
Hure 2008 [78]	n/a	284.4 mcg	n/a	n/a	n/a	no	All
Jun 2020 [79]	n/a	n/a	n/a	787 mcg DFE	n/a	yes, no food.	All
Livock 2017 [80]	n/a	n/a	T1: 247 mcg	T1: 522 mcg	T1: 840 mcg	yes	
			T2: 245 mcg	T2: 527 mcg	T2: 760 mcg		
			T3: 251 mcg	T3: 518 mcg	T3: 690 mcg		<19
Martinussen 2012 [92]	n/a	n/a	n/a	month 1 402 mcg month 2 605 mcg month 3 676 mcg	n/a	yes, no Food	<24
Masih 2015 [59]	T1: 313 ± 140 mcg DFE T3: 297 ± 131 mcg DFE	T1: 483 mcg DFE T3: 465 mcg DFE	T1: 96 ± 54 mcg T3: 96 ± 50 mcg	1000 mcg	n/a	yes	<16
Murphy 2021 [51]	148.2 mcg DFE	346.7 mcg 512.5 mcg DFE	226.6 mcg	1100 mcg	n/a	yes	24–26
Murphy/Rose 2021 [50]	140 mcg DFE	485 mcg DFE	333 mcg DFE	1000 mcg	2167 mcg DFE	yes	8–16 weeks
Pick 2005 [88]	-	331 mcg	n/a	n/a	n/a	no	20–38
Plumptre 2015 [62]	T1: 483 ± 203 mcg DFE T3: 465 ± 186 mcg DFE	n/a	n/a	1000 mcg	n/a	yes	10–22 weeks
Roy 2012 [93]	n/a	473 mcg DFE	n/a	1338 mcg	1811 mcg DFE	yes	10–22 weeks
Shin 2016 [83]	n/a	627.6 mcg DFE	n/a	781.8, 1329 mcg DFE	n/a	yes	All
Trivedi 2018 [52]	n/a	n/a	n/a	n/a	T1: 930 mcg T2: 1238 mcg	yes	<25 weeks
Whitrow 2009 [66]	T1: 224.7 mcg T3: 208.4 mcg	n/a	n/a	T1: 658.3 mcg T3: 300 mcg	n/a	yes	<16

DFE = dietary folate equivalents; n/a = not applicable, did not measure; WD = Western diet; mcg = micrograms; T1: Trimester 1; T2: Trimester 2; T3: Trimester 3.

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
