# Peer review of "Women Taking a Folic Acid Supplement in Countries with Mandatory Food Fortification Programs May Be Exceeding the Upper Tolerable Limit of Folic Acid: A Systematic Review"

_nutrients, 2022, doi:10.3390/nu14132715_

Round 1

Reviewer 1 Report

The authors present a complete review of the literature related to folate supplementation in women that might conceive or in pregnant women. The review criteria are clearly outlined, and the sample size of papers is significant. There are mosty grammar errors to correct, or what appears to be formatting errors associated with the template. 

In the abstract:

"This reviews aim" should be "review's".

Define the mono- versus polyglutamate forms of folate. Actual chemical structures would be a helpful addition. 

p3. line 123 Unneeded space sends period to next line.

p.5 lines 204-205. There seems to be a list of issues, but it stops at 2. An and would be a better conjunction. 

Lines 270-276 have serious formatting concerns.

Line 305, "were" instead of "wee"

Line 358-359 The restatement of the meaning of DFE is unnecessary.

375-376 Please restate the sentence as the logic is unclear.

383-387 Please restate with a period included in the text somewhere, as the sentence is so long that it becomes very hard to parse

410

418

423

431-432 all have similar formatting errors

Author Response

Dear Reviewer 1, please find attached response to your comments.

thank you very much for reviewing this paper.

regards carolyn

Reviewer 2 Report

Excessive intake of FA (folic acid) causes UMFA appearing in the body, which has undesirable biological effects. For this reason, an upper intake limit has been established for FAs. This article presents the systematic review of FA intake and clearly shows excessive intake in high income countries, suggesting that many pregnant women intake of higher amount of  FAs through supplements. So this article is impressive for public health.

 The followings are my comments.

1)The subject of this paper is to analyze folic acid and folate intake in high income countries. Folate and folic acid  should be distinguished. Folic acid should therefore be referred to as FA and folate as Folate in this paper. 

3)Many abbreviations are used without any description in some words. This makes this paper difficult to be understood. All abbreviations should be summarized in a footnote. 

4)One carbon metabolism (folate metabolism) is explained in detail., but there is no metabolic map, what makes it difficult to be understood. For this reason, the metabolic pathway of folateand folic acid should be illustrated ,including related enzymes. Figure 1, which shows the pathway of absorption, is a good diagram. 

5)This article shows that there are some pregnant women who intake higher amount of FA and undesirable UMFA is detected in their blood. However, it is argued that forinic acid and MTHF are safer alternatives to FA. This is a important beneficial news for pregnant women. This might be an extremely useful concept. However, it is unclear whether the high dose of 800 mcg of MTHF is safe, even though it is said to have no adverse effects.

In pregnant animals, FA and MTHF administration was compared.  It is reported that MTHF altered the gene expression in the pituitary gland of the offspring, causing the offspring to have an increased appetite and develop insulin resistance. Therefore, it is unclear whether HTMF is safer than FA. Please clarify your new proposal. 

6)About DFE(l-269, l-126Box-1:

The formula for calculating DFE is given, but the authors say this is not accurate. This formula is based on a small number of people, and its accuracy is questionable (l-347-361). If the .DFE conversion formula is now questionable, it is inappropriate as a formula for calculating DFE and should be deleted in this paper, and box-1 should not be shown. In part of Table 1, the DEF value is shown, but this value should be differentiated into  Folate and FA.

7)The following text should indicate the value of the RDA. ofb each country.

(l-249-L-291) One study that investigated only food sources of folate found that 80% of women of 249 reproductive age did not meet the recommended daily allowance (RDA) [90], while 250 another study showed that only 23% of women achieved the RDA [86]. 

8)The different EAR values for each country should be stated. 

9)In Table 1, the following sentences need to be corrected. They are given as examples. Similar sentences, which  should be corrected,  are scattered. The author should check and correct other parts of the texts..

① Gakins 2012: In500mcg achieved without supplementation ”、

500 mcg should be to 50 %

② Gaskins 2019 In 20% women consumed 1000mcg of supplement folate

  →Folate should be to FA

③ Pick 2005  In “ food does not achieve FA 400g/day ”

FA g/day should be to  400mcg/day

③ Rai 2014 In Most women achieved minimum but  few reached 906 nmol/L

  Most women achieved minimum but a few reached 906 nmol/L (?)

④ In At 1000mcg/d, less than 25% of women had rbc folate level that prevent NTD’s ”

At 1000mcg/d, less than 25% of women had RBC folate level that prevent NTD’s

Author Response

Dear Reviewer 2,

Please find attached a reply to your comments.

Thank you for reviewing my paper.

Regards

carolyn
